# Inducing Angiogenesis in the Nucleus Pulposus

**DOI:** 10.3390/cells12202488

**Published:** 2023-10-19

**Authors:** Sheela R. Damle, Agata K. Krzyzanowska, Maximilian K. Korsun, Kyle W. Morse, Susannah Gilbert, Han Jo Kim, Oheneba Boachie-Adjei, Bernard A. Rawlins, Marjolein C. H. van der Meulen, Matthew B. Greenblatt, Chisa Hidaka, Matthew E. Cunningham

**Affiliations:** 1HSS Research Institute, Hospital for Special Surgery, 515 E 71st Street, New York, NY 10021, USA; 2Weill Cornell Medical College, Cornell University, New York, NY 10065, USA; 3Meinig School of Biomedical Engineering and Sibley School of Mechanical & Aerospace Engineering, Cornell University, Ithaca, NY 14853, USA; 4Department of Genetic Medicine and Belfer Gene Therapy Core Facility, Weill Medical College of Cornell University, New York, NY 10065, USA

**Keywords:** intervertebral disc, angiogenesis, osteogenesis, fusion, gene delivery, proteoglycanase nucleus pulposus

## Abstract

Bone morphogenetic protein (BMP) gene delivery to Lewis rat lumbar intervertebral discs (IVDs) drives bone formation anterior and external to the IVD, suggesting the IVD is inhospitable to osteogenesis. This study was designed to determine if IVD destruction with a proteoglycanase, and/or generating an IVD blood supply by gene delivery of an angiogenic growth factor, could render the IVD permissive to intra-discal BMP-driven osteogenesis and fusion. Surgical intra-discal delivery of naïve or gene-programmed cells (BMP2/BMP7 co-expressing or VEGF_165_ expressing) +/- purified chondroitinase-ABC (chABC) in all permutations was performed between lumbar 4/5 and L5/6 vertebrae, and radiographic, histology, and biomechanics endpoints were collected. Follow-up anti-sFlt Western blotting was performed. BMP and VEGF/BMP treatments had the highest stiffness, bone production and fusion. Bone was induced anterior to the IVD, and was not intra-discal from any treatment. chABC impaired BMP-driven osteogenesis, decreased histological staining for IVD proteoglycans, and made the IVD permissive to angiogenesis. A soluble fragment of VEGF Receptor-1 (sFlt) was liberated from the IVD matrix by incubation with chABC, suggesting dysregulation of the sFlt matrix attachment is a possible mechanism for the chABC-mediated IVD angiogenesis we observed. Based on these results, the IVD can be manipulated to foster vascular invasion, and by extension, possibly osteogenesis.

## 1. Introduction

Patients experiencing intolerable axial spine pain refractory to non-operative treatment, either in the setting of *spinal instability* (hypermobility due to fracture, spondylolisthesis, or end-stage spondylosis), regional *spinal deformity* (scoliosis, kyphosis), or certain *spinal infections/neoplastic* diagnoses, are frequently indicated for segmental spinal fusions as definitive treatment. Fusion of the affected spinal segments limits or eliminates axial pain by stabilizing treated hypermobile segments, by correcting current and preventing progressive spinal deformity, and by augmenting structural integrity of the spine in situations of impending pathological fractures. The current surgical technique includes using metal implants to immobilize the spinal vertebrae to be fused, decortication of strategic surfaces of the vertebrae to increase osteoprogenitor cell populations in the intended fusion site and to induce local osteogenesis mechanisms, and placement of bone grafts to optimize osteoconduction and osteoinduction. The expectation is that a non-mineralized osteoid anlage is generated, the anlage evolves and undergoes mineralization forming a mature bone fusion mass, and that as the anlage evolves/mineralizes, the spine progressively stiffens until no motion is detectable, resulting in mechanical spinal fusion. “Open technique” spinal fusions are long and taxing surgeries, exposing patients to complications and challenging post-operative convalescence [1,2]. To minimize surgical morbidity, minimally invasive (MI) spinal fusion techniques have developed, with their promise for less surgical trauma, shorter hospital stays, quicker recoveries and fewer peri- and post-operative complications [3,4]. Taken to an extreme, MI spine fusion could be an injection-based percutaneous treatment that induces heterotopic ossification sufficient to mechanically stabilize the intended spinal segment(s), possibly eliminating surgery, recovery, and major complications all together. Percutaneous or relative-MI delivery of a variety of treatments in comparative models to induce posterior [5,6] and anterior [7,8,9,10] fusions offer hope that percutaneous fusion may be possible for clinical use one day.

We previously delivered bone morphogenetic proteins (BMPs) to endplate-punctured but otherwise intact lumbar intervertebral discs (IVDs) in Lewis rats, and observed that, although we could induce bone anterior to the IVD, the nucleus pulposus (NP) compartment of the IVD was well preserved and was remarkably resistant to osteogenesis [7]. We used non-invasive induced angular displacement (NIAD) at 4-week intervals to measure the loss of spinal motion over time, and showed the pre-operative NIAD measurement decreased in all surgical groups (attributed to soft tissue scarring) by ~25%, but only the heterodimer BMP group showed progressive NIAD losses beyond 4 weeks. BMP-induced bone formation was dose-responsive to the relative osteoinductiveness of the treatment delivered (heterodimer (BMP2/BMP7) > homodimers (BMP2 or BMP7 or mixed BMP2 + BMP7) > negative control (betagalactosidase)), but significant spinal fusion was only observed for the heterodimer-treated group. Progressive loss of NIAD over the course of the experiment predicted relative bone production, increased stiffness assessed by 4-point bend biomechanics, and spinal fusion (assessed by palpation and radiographic endpoints). Our findings supported the opinion that the NP is a very inhospitable location to induce bone formation, either due to the known IVD avascularity and hypoxia, or other non-vascularity-related mechanisms that make the NP a barrier to neo-osteogenesis.

Considering the disc as a physical or chemical barrier to bone induction, we hypothesized that a treatment to remove the IVD matrix could render the disc better able to support bone induction for fusion. Chymopapain is perhaps the most familiar of the enzymes used experimentally and clinically as a chemonucleolytic agent (a substance able to dissolve disc tissue) [11,12]. A second-generation agent, chondroitinase ABC (chABC), is safer than chymopapain in comparative models including hamsters, rabbits and pigs [13,14,15,16]. Previously, application of chABC to (Sprague Dawley) rat IVDs decreased proteoglycan content and altered spine and IVD mechanics [17,18]. Due to the safety profile, demonstrated experimental utility, and commercial availability, chABC was selected to chemically degrade the IVD in our rat anterior spine fusion model. 

How avascularity is normally maintained in the NP, and therefore how it might be manipulated, is poorly understood despite the multiple molecules and mechanisms that have been suggested, including tissue inhibitor of metalloproteinase-3 (TIMP3) [19], semaphorin 3A [20], thrombospondin-1(TSP1) [21], TSP2 [21], TSP3 [21] and TSP5/collagen oligomatrix protein (COMP) [22], or chondromodulin-1 (CHM1) [23] and the related molecule tenomodulin (TNMD) [24]. Furthermore, when the expression of these molecules was manipulated and IVD angiogenesis assessed, angiogenic vessel penetration in the annulus fibrosus (AF) was observed, but the NP remained avascular [20,25,26,27], suggesting *unique* or *redundant* mechanisms for angiogenesis control for different parts of the IVD (NP, AF and cartilage endplate (CEP)) [27,28]. To make the angiogenesis mechanism even more complex are the more recent findings that resident NP cells express vascular endothelial growth factor-A (hereafter, VEGF) [29,30,31,32,33], with VEGF expression being upregulated by IVD hypoxia [31,34,35] and degeneration [29,30,35], downstream of hypoxia-inducible factors (HIF)-1 [36,37,38] and -2 [34], and NP-expressed VEGF appears to act in a paracrine/autocrine manner as a survival factor for NP cells [31,36,37,38]. Therefore, a thorough description of IVD anti-angiogenesis would need to explain how the AF, CEP and NP repel vascular ingrowth and how the NP-expressed VEGF is sequestered from acting on angiogenesis-sensitive tissues external to the disc space. We hypothesized that generating a blood supply into the disc would allow BMP-driven bone formation within the IVD, and that angiogenesis inside the IVD could be achieved through VEGF gene delivery into the NP that would overpower the anti-angiogenesis mechanisms.

Osteoinductive signaling through BMP is up-regulated in response to VEGF treatment [39], and reciprocally, VEGF signaling is up-regulated by BMP treatment [40], with both acting as chemotaxis signals for osteoblasts and endothelial cells [41]. Co-treatment with VEGF and BMP produces bone significantly earlier and to a greater extent than BMP alone, and VEGF inhibition impairs BMP-induced osteogenesis [42,43]. Interestingly the ratio of VEGF-to-BMP delivered affected bone induction, with optimal osteogenesis observed with VEGF-to-BMP at a 1:5 ratio [42,43]. This introduces our third hypothesis: co-delivering BMPs and VEGF would augment osteogenesis and fusion in our model. 

Here, we hypothesize that (1) intra-discal delivery of cells genetically modified to express VEGF_165_ will overpower the anti-angiogenesis homeostasis mechanisms of the IVD and drive angiogenesis within the IVDs prepared by endplate perforation, making the IVDs permissive for bone induction by BMP, (2) co-delivery of BMP-expressing cells with VEGF-expressing cells will additively or synergistically enhance bone induction and spinal fusion, and (3) chondroitinase ABC delivery will destroy IVD chondroitin/dermatan sulfate aminosaccharide proteoglycans, disrupting the NP inhibition of neo-osteogenesis, and make the IVD permissive to BMP-induced bone formation and spinal fusion.

## 2. Materials and Methods

### 2.1. Preparing Treatments and Aminal Surgeries

Bone marrow mesenchymal stromal cell (BMSC) cultures, virus stocks, and transgene-expressing BMSCs for implantation were generated as previously described [7,44,45]. All animal surgeries and primary cell cultures were performed under a Hospital for Special Surgery (HSS) Institutional Animal Care and Use Committee (IACUC) approved protocol (#03-08-06R), and in compliance with ARRIVE guidelines and HSS Center for Lab Animals Services (CLAS) guidelines and regulations. BMSCs from *n* = 20 male approximately 8-week-old (200–250 g) Lewis rats were expanded in monolayers, transduced 18–24 h prior to surgery with 10^5^ particle units (pu)/cell with recombinant adenoviral (Ad) vectors encoding human bone morphogenetic protein-2 (AdBMP-2), human BMP-7 (AdBMP-7) and human VEFG_165_ (AdVEGF) transgenes, and a sample of the gene-programmed cells was assessed 3 days after transduction for transgene expression using ELISA (VEGF and BMP2 using kits from R&D Systems Inc., Minneapolis, MN, and BMP7 using a kit from Alpha Diagnostic International, San Antonio, TX, USA). When representative aliquots of cells demonstrated lack of intended gene expression, implanted animals were immediately euthanized (*n* = 59).

Experimental groups were designed to receive equivalent treatment dosages to allow direct comparisons between groups. Just prior to implantation, cells were trypsinized, concentrated by centrifugation, counted by hemocytometer, and suspended in growth media into a slurry so that 10^6^ cells would be delivered in 25 µL, a previously optimized dosage [44]. Chondroitinase-ABC (chABC, Sigma, Burlington, MA, USA cat# C2905) was used at a dose of 25–30 milli-Units. A non-operated group (Mock) and an operated group implanted with non-Ad-infected cells (Naïve) were included as negative controls. Operated treatment groups were chABC (implanted with chABC and 6 parts naïve cells), VEGF (implanted with 1 part AdVEGF-infected cells and 5 parts naïve cells), BMP (implanted with 1 part naïve cells and 5 parts AdBMP2 and AdBMP7 doubly transduced cells), and all combinations of these three treatments, with the group size (*n* ≥ 15 animals) determined by power analysis for the primary outcome of palpation fusion.

Treatments were implanted into the L4–L6 disc spaces. Approximately 8-week-old (200–250 g) male Lewis rats (*n* = 221) were anesthetized, sterilely prepared and subjected to a transperitoneal exposure of lumbar levels L4–L6, as previously described [7,44,45]. Vertebral endplates were punctured using a 22 Ga needle passed through the anterior L5 body proximally into the L4 body (through the L4/5 disc), and repeated directed distally into the L6 body (through the L5/6 disc); 2–3 passages per disc were performed. Using a 25 Ga needle, 10^6^ genetically modified BMSCs were delivered to L4/5 and L5/6 disc spaces, and wounds were closed in layers. Animals were recovered, and allowed ad libitum food, water and activity. Morbidity and mortality data have been reported separately [45].

### 2.2. Noninvasive Induced Angular Displacement Assessments

High-definition digital Faxitron radiograph images were used to monitor spinal noninvasive induced angular displacement (NIAD) throughout the time course as we have described [7,46]. Pre-operatively (*n* = 216) and at 4, 8, and 12 weeks (*n* = 136) after surgery, NIAD quantification was performed under isoflurane anesthesia, with the animals positioned supine in a custom 90-degree bending bracket (in both right and left bending positions), segmental angles were measured for each level L4–S1, and data are reported as the sum of the right and left angles for each level (total coronal plane displacement). A subset of the images was measured by a second blinded investigator to measure inter-observer reliability using the Interclass Correlation Coefficient (ICC). NIAD data were normalized to the time-zero treatment group means to correct for small groupwise differences in the baseline assessments (Figure A1 in Appendix A), and are presented as ‘Percent of Pre-OP’ in the manuscript.

NIAD assessment at 12 weeks was also adapted as a fusion prediction technique, with fusion defined as segmental measurements that were at least 3 standard deviations below the mean values of the preoperative total population (*n* = 216). For the L4/5 level, the average was 21.6° and the standard deviation was 2.6°, making post-operative specimens considered as “fused” when their NIAD measurement was under 13.8°. For L5/6 the average ± St Dev was 18.9° ± 3.5°, with “fused” defined as under 8.4°, and for L4–6, it was 37.7° ± 4.4°, with “fused” defined as under 24.5°. These “NIAD fusion” critical values would be expected to identify the lowest 0.15% of “normal” motion in the pre-operative cohort, and none of the pre-operative animals met these criteria for “NIAD fusion”.

### 2.3. Bone Formation and Fusion Assessments

Euthanasia, post-mortem radiography, and radiographic bone induction/fusion was performed as we have described [7]. Animals were euthanized at 12 weeks by CO_2_ inhalation followed by cervical dislocation. AP and lateral Faxitron images were used to evaluate for unintended heterotopic bone production in the thorax and abdomen, but none was noted. The lateral Faxitron images were also used to score bone induced and probability of fusion of the L4/5 and L5/6 levels. *Graded bone formation* and *fusion likelihood* were assessed on lateral radiographs by 3 blinded investigators for each level using a 0–2 scale: 0 assigned to minimal-to-low-level bone production and suspected non-fusion status, 1 assigned to intermediate bone production and possible fusion status, and 2 assigned to a large amount of bone production and highly likely fusion status. Rater grades were summed for each level independently, and interpreted as follows: 0–2 as minimal bone production (not fused), 3–4 moderate bone production (possible fusion), and 5–6 abundant bone production (fused). Sums were used to compare groups for treatment effects, and dichotomous rendered data (5–6 was fused, 0–4 was not fused) were used for comparisons of fusion assessment methods. *Categorical* radiographic fusion (presence/absence of bridging bone on lateral image) was performed by 3 raters independently, with sample status assigned by the majority.

Spines from euthanized animals were recovered, and palpation fusion assessment was performed as previously described [7]. Spines from L3-S1 were explanted and stripped of soft tissues other than near the intervertebral discs and induced fusion bone. Manual palpation has been reported as the “gold standard” for rat spine fusion assessment [47,48], and was considered as the ground truth for sensitivity, specificity and positive/negative predictive values reported. All spines were palpated by 3 blinded observers for fusion at L4/5 and L5/6, and were graded as “fused” = no motion, or “not fused” = motion detected; score sums from 3 raters were used to statistically test treatment effects, and dichotomous fusion status by level was assigned by majority for reporting fusion success and correlation testing. 

### 2.4. Micro-CT/Histology and Biomechanics Assessments

Spines were prepared for micro-CT and histology or in vitro biomechanical testing as previously described [7]. Samples were either stored frozen (−80 °C) for mechanical testing (*n* ≥ 10/group) or were fixed for 48 h with 4% paraformaldehyde in codylate buffer for micro-CT and histological analysis (*n* = 5 or 6 per treatment group). Quantitative micro-CT scans of rat spines from L3–S1 were digitally reconstructed, processed using GE MicroView software (latest version 2.1) (GE Healthcare, Chicago, IL, USA), and quantified with the mineral threshold set at 1500. Measurements obtained were total volume fusion bone induced for the combined L4–6 segment, and bone formed within the L4/5 and L5/6 discs. Following micro-CT, samples were processed for histology with decalcification in EDTA, paraffin embedding, sagittal sectioning at 5 µm, and staining with H&E (general morphology), Alcian Blue (cartilage), Picrosirus Red (collagen), and immunostained for VE-cadherin (BV9, Santa Cruz Biotechnology Inc, Dallas, TX) with selected samples undergoing follow up immunostaining with CD31 (ab182981, Abcam PLC, Waltham, MA). Mid-sagittal sections were chosen for scoring, samples were re-sectioned or re-embedded to obtain optimal appearance, and histological assessments of a disc were excluded unless one or both endplates showed a surgical puncture site from the disc preparation (except the Mock group where midsagittal was estimated). The L4/5 and L5/6 levels were assessed and scored independently, making possible specimens per group *n* = 10–12, but with midsagittal requirement applied, the actual numbers were *n* = 11 (chondroitinase ABC (chABC)/VEGF), *n* = 10 (Mock, Naïve, BMP, chABC/BMP), *n* = 9 (VEGF, BMP/VEGF, chABC/BMP/VEGF), and *n* = 6 (chABC). Histology samples were assessed for IVD damage, induced bone and intra-discal vascularity using a rubric (Table A1) adapted in part from prior studies [49,50,51]. The final score for damage of a sample (0–8) was the sum of 4 sub-scores (NP damage (0–2), AF damage (0–2), alterations in interfaces (0–2) and Alcian Blue staining (0–2)); scores for bone formation (0–3) and angiogenesis (0–2) did not require summation.

Biomechanics specimens were potted and subjected to four-point bending mechanical testing, in vitro angular displacement (IVAD), and four-point load to failure in extension, all as previously described [7,46]. Proximal (L3 and proximal L4) and distal (distal L6 and S1) ends were potted in acrylic bone cement (COE Tray Plastic, GC America, Chicago, IL, USA) in custom rectangular aluminum fixtures, and four-point bending (outer span 50 mm, inner 22.5 mm) was performed over the combined L4–L6 segment with loads applied at 0.5 N/sec up to a maximum of 4 N (ELF 3200, EnduraTec, Eden Prairie, MN, USA). Specimens were assigned numbers, with numbers randomly chosen for processing, and each specimen was tested in random order for right and left lateral bending, and extension and flexion by rotation of the rectangular blocks. IVAD was measured from a digital photograph of the four-point bending apparatus during maximal loading in the 5th cycle for each direction. Four-point-bending moment-to-failure destructive testing was performed in extension, with failure location (L4/5 or L5/6 disc) and moment-at-failure recorded.

### 2.5. NP Cell and Disc Organ Preparation

Bovine NP (bNP) cell cultures and rabbit disc organs were prepared as previously described [52]. bNP cells were obtained from three cadaveric bovine tails of young adult animals following approval of the HSS IACUC, and in compliance with ARRIVE guidelines and HSS CLAS guidelines and regulations. Discs were dissected, endplates removed, and NP tissue was mechanically and enzymatically extracted. Cells were washed and seeded at 2.8 × 10^4^ cells/cm^2^ density in complete medium (high-glucose Dulbecco’s Modified Eagle Media (DMEM; Gibco, Grand Island, NY, USA), 10% Fetal Bovine Serum (FBS; Gibco), 1% antibiotic-antimycotic (Gibco) and 10 µM HEPES buffer (Gibco)). bNP cells were incubated in a humidified atmosphere of 5% CO_2_ at 37 °C, in either normoxic (not manipulated, 21%) or hypoxic (2% oxygen controlled by a BioSpherix C-Chamber inserted into the same incubator; BioSpherix, Lacona, NY, USA). Cells were used for experiments at passages 2–10, with careful monitoring of cell morphology changes with light microscopy at media changes (Figure A2). New Zealand White (NZW) rabbit disc organs were isolated from young adult animals, following approvals by the HSS IACUC, and in compliance with ARRIVE guidelines and HSS CLAS guidelines and regulations. Spines were obtained after the animals had been euthanized and discarded by independent investigators performing non-spine-related protocols. L1–S1 spine segments were excised, posterior elements and anterior soft tissues were debrided, and lumbar vertebral–disc complexes were recovered allowing isolation of IVDs and NP tissue, with *n* = 4 IVDs obtained from each of *n* = 8 rabbits.

### 2.6. Western Blotting Experiments

Western immunoblotting of soluble VEGF-R1 (sFlt) was performed using bNP cell cultures and NZW IVD tissue. bNP cells at near confluence had conditioned media collected one day after feeding, the cell monolayer was washed 3 times with ice cold PBS, and the cells and matrix were collected in lysis buffer (10% glycerol, 1% Triton X-100, 50 mM Tris pH 7.5, 150 mM NaCl, 1 mM EDTA, 10 mM NaF, 2 mM Na_3_VO_4_, 1 mM 1,10-phenanthroline, 4 mM PMSF, and 1x protease inhibitor cocktail (Roche, Basel, Switzerland)). The lysate was incubated on ice for 15 min, clarified by centrifugation (14,000× *g* for 10 min), supernatants were normalized by protein, and prepared for SDS PAGE using 4x Laemmli buffer (Bio-Rad, Hercules, CA, USA). Paired conditioned media samples were normalized by volume and incubated overnight at 4 °C with Concanavalin A-sepharose 4B beads (GE Healthcare, Piscataway, NJ, USA) to enrich N-linked glycoproteins (including sFlt), beads were pelleted, and samples were prepared for SDS PAGE using 1× Laemmli buffer. SDS PAGE, transfer to nitroceullulose membrane, blocking, probing with anti-VEGF-R1 (V4262, Sigma-Aldrich, Burlington, MA, USA) and image capture with SuperSignal ECL (ThermoFisher, Bohemia, NY, USA) were similar to our prior description [53,54]. NZW IVDs were opened with a scalpel and NP contents from 2 discs were pooled into microtubes, suspended in sterile TBS and stored on ice until digestions were performed. Digestions (500 µL) were performed in duplicate for 1, 2 or 4 h, or overnight, in either 0.5% collagenase type II (Worthington Biochemical Corporation, Lakewood, NJ, USA), or 200 mU/mL chondroitinase ABC (C2905, Sigma-Aldrich, St. Louis, MO, USA), suspended in DMEM with 1× antibiotic/antimycotic at 37 °C and shaking at 225 rpm. Collagenase type II was chosen as a control for matrix digestion due to its use in generating NP cell primary cultures (suggesting NP cell tolerance of its activity while it disintegrates the IVD matrix), its description as a mixed *protease* activity (it digests collagen and other matrix proteins), and to demonstrate the difference between matrix *proteoglycan* destruction (chABC) and matrix *protein* destruction. After digestion, cells and debris were clarified by centrifugation (14,000× *g* for 10 min), supernatants were normalized for protein, and soluble N-linked glycoproteins were concentrated using Concanavalin A-sepharose beads. Beads were then pelleted, prepared for SDS PAGE with 1× Laemmeli buffer, size separated on 10% gels, transferred to nitrocellulose, blocked, probed with anti-VEGF-R1 and imaged with SuperSignal ECL.

### 2.7. Statistical Testing and Analysis

Secondary to non-uniform fusion bone induction and consequent non-gaussian experimental spine fusion in the most successful groups, measurement data (NIAD, multimodal fusion, bimodal bone formation, and fusion stiffness) did not meet normal data distribution assumptions of parametric tests, and data were compared using the non-parametric Independent Samples Kruskal–Wallis test (ISK-W), with Dunn–Bonferroni post hoc pairwise testing using SPSS (v.22, SPSS Inc., Chicago, IL, USA), as per our prior reports [7,46]. NIAD rater reliability was measured using ICC (SPSS), and time effects on NIAD assessment was measured by ISK-W after data were stratified by treatment and dependent groups defined by time of assessment (0, 4, 8 or 12 weeks). Fusion data were compared through the sum of three raters for palpation (range 0–3) and for categorical radiographic fusion (range 0–3), as was faxitron bone formation data (range 0–6). Inter-rater agreement for dichotomous/categorical data (multimodal fusion assessments) was measured by Fleiss’ Kappa (www.statology.org>fleiss-kappa-excel (accessed on 15 May 2023)). Agreement was interpreted for ICC as *poor* for <0.5, *moderate* for 0.5 to 0.75, *good* for 0.75 to 0.9, and *excellent* for > 0.9 [55]; Fleiss’ Kappa was interpreted as *poor to fair* for <0.4, *moderate* 0.41 to 0.6, *substantial* for 0.61 to 0.8 and *almost perfect* for 0.81 to 1.0 [56]. Data correlation was performed using Pearson’s r for continuous data comparisons and with Spearman’s *ρ* when one or both data sets were non-continuous (SPSS). Testing of treatment effects for graded fusion data (rendered to fused/not), and of prior fusion data [7] to current data, were performed using the Fisher exact test (GraphPad Prism 9, GraphPad Software, San Diego, CA, USA). The four-point biomechanics failure in extension locations was compared using the One-sample *t*-test for Proportion (https://www.medcalc.org/calc/test_one_proportion.php (accessed on 18 May 2023)). Summarized data were reported using boxplots (median as a centralizing line, 25th and 75th percentiles as box limits and whiskers representing maximum and minimum data points), histograms, stacked bar graphs, or violin plots (similar to boxplots but directly illustrates the data distribution for ordinal data), and all graphs were generated using GraphPad Prism 9.

## 3. Results

### 3.1. BMP and BMP/VEGF Decreased NIAD over a 12-Week Time Course

Compared to the pre-operative baseline, operated groups at 4 weeks showed NIAD decreases for L4/5 and L5/6 (*p* < 0.001 for each level, Figure A3 and Figure A4) and for the L4–6 combined segment (*p* < 0.001, Figure 1 and Figure A5). The NIAD decrease in the operative groups was thought to reflect non-specific surgical scarring (Naïve, VEGF), exaggerated scarring due to IVD proteoglycan destruction (chondroitinase ABC (chABC), and combinations), or possibly the presence of induced bone (BMP and combinations). The BMP and BMP/VEGF treatment groups were most frequently identified as being significantly decreased for each of the segments. The L6/S1 level showed NIAD increases of 3–30% in operated groups (*p* < 0.0001), but did not show treatment effects between the operated groups (Figure A6). At weeks 8 and 12, the BMP and BMP/VEGF treatment effects for some of the intergroup comparisons in L4/5 and L4–6 were diminished or lost, despite significant differences being found for each segment for both time points overall (each with *p* < 0.001). This result was attributed to gradual progressive NIAD decreases after 4 weeks, such as those shown for L4–6 chABC (Figure 1), and to more data dispersion in the BMP and BMP/VEGF measurements at 8 and 12 weeks (Figure 1 and Figure A5). Comparing operated groups for the L5/6 and L6/S1 levels also showed some gains and losses of significant treatment effects over time, but BMP/VEGF was most frequently identified as the different treatment (Figure A4 and Figure A6). The lack of more significant NIAD attenuation for the chABC/VEGF/BMP and chABC/BMP groups was surprising, as chABC, BMP and VEGF/BMP treatment all caused significant attenuations, but when combined, there was (or trended towards) less NIAD attenuation, as if the combination of chABC with BMP or VEGF/BMP were antagonizing one another (Figure 1, Figure A3, Figure A4, Figure A5 and Figure A6). When stratified by treatment to allow testing of the effect of time, NIAD data (L4/5, L5/6, and L4–6) showed significant differences for each level of each operative treatment (*p* ≤ 0.001), with the exception of naïve cell implantation at L5/6 (*p* = 0.308). The most consistent differences were between pre-operative to any post-operative assessment of a treatment, with only chABC treatment demonstrating progressive loss of NIAD after 4 weeks for the L4/5 (*p* = 0.045 to 0.002) and L4/6 (*p* = 0.017 to 0.02) segments (Figure 1). ICC inter-rater reliability for the NIAD data was *good* at 0.884 (95%CI 0.869 to 0.897, *n* = 1072, *p* < 0.001). No differences were noted comparing the current 12-week NIAD results to our prior published findings (Table A2).

### 3.2. Mechanical Stiffness Testing Parallels NIAD Findings and Independently Confirms BMP- and BMP/VEGF-Dependent Increased Spinal Stiffness

IVAD assessment for the combined L4–6 segment showed significant treatment effects between groups in each simple (left, right, flexion and extension) and combined (coronal and sagittal) direction assessed (*p* < 0.001, Figure 2 and Figure A7). Treatment effects were most numerous in the right and coronal comparisons, and the BMP and BMP/VEGF treatments stiffened the spines more predictably than others. Presumably secondary to inducing soft tissue scarring in or around the IVD, chondroitinase ABC (chABC) treatment caused significant stiffening in the coronal (Figure 2) and left-bending (Figure A7) IVAD directions, but as was found for NIAD, this chABC-induced IVAD stiffness was not augmented by combination with BMP or BMP/VEGF treatments (Figure 2 and Figure A7). BMP- or BMP/VEGF-induced stiffness had a similar antagonistic effect when combined with chABC. Non-destructive four-point bending demonstrated significant differences only in left (*p* = 0.003) and right (*p* < 0.001) bending (Figure 2), but not in flexion (*p* = 0.136) or extension (*p* = 0.211) (Figure A7), and only BMP and BMP/VEGF treatments had significant inter-group differences. Load to failure in extension was affected by treatment (*p* = 0.031), but the only significant intergroup difference detected was between Naïve and chABC treatments (*p* = 0.007). Spine failure location (L4/5 or L5/6 level) was dictated by the fusion status of the level for fused specimens (*n* = 14), was not determinable in some cases (*n* = 4), and these samples (*n* = 18) were excluded from the failure site comparison. Consistent with our prior work [7], spine failure was more likely to occur at L5/6 (52/72) than L4/5 (20/72) (*p* < 0.001, z = 3.767, 95%CI: 60.39% to 82.12%), suggesting that L4/5 and L5/6 differ with regard to mechanics or healing potential. The IVAD and NIAD correlated significantly (r = 0.788, *p* = 0.01, *n* = 90), consistent with our prior study [7]. No differences were noted comparing our prior results to the current biomechanical stiffness data or to IVAD for samples receiving BMP treatment; however, small but significant differences were found for IVAD-negative controls (Naïve versus Ad-LacZ expressing), potentially due to a subtle effect of the Adenoviral vector and altered data distributions in the datasets (Table A2).

### 3.3. BMP and BMP/VEGF Treatment Resulted in Increased Frequency of Spinal Fusion Assessed at 12 Weeks

Manual palpation testing was consistently positive in only in the BMP and BMP/VEGF treatment groups (Figure 3A). BMP and BMP/VEGF fusion rates did not differ by level for each treatment (*p* ≥ 0.544) or by treatment at each level (*p* = 1). Although chondroitinase ABC (chABC) treatment did not demonstrate an ability to drive fusions when assessed by palpation (Figure 3A) or by radiographic (Figure 3B,C) assessments, when chABC was combined with BMP or VEGF/BMP, it antagonized the ability of BMP or VEGF/BMP to drive spinal fusions, in the same manner we observed for NIAD and IVAD. Categorical (Figure 3B) and Graded (Figure 3C) radiographic fusion also occurred most frequently in the BMP and BMP/VEGF groups. No differences were noted between the current findings and our prior published results for palpation, or categorical or graded fusions (Table A2). Fusion success measured by ‘critical’ NIAD values at 12 weeks was similar to the other fusion modalities for BMP and BMP/VEGF (Figure 3D), but additional treatment groups also met this ‘NIAD fusion’ definition, resulting in a worse positive predictive value (PPV 44.1%) than graded fusion (PPV 92%) and categorical fusion (PPV 84.4%) when palpation success was used as the ground truth. The ‘critical’ NIAD measurement appeared to be a better assessment of stiffness than actual fusion. Interestingly, ‘critical’ value NIAD illustrated a trend for treatment interaction, decreasing the stiffness generated by chABC alone when combined with VEGF, BMP or VEGF/BMP, and showing statistically significant antagonism for the L4–6 combined segment comparing BMP or VEGF/BMP ± chABC (Figure 3D). Agreement between observers was *almost perfect* for palpation (Fleiss’ K = 0.869) and categorical fusion (K = 0.818), and was *substantial* for graded fusion (K = 0.709).

### 3.4. Decreased Mobility and Increased Spinal Fusion Was the Result of Bone Formation around the L4/5 and L5/6 IVDs, but Not Bone Production Inside the IVDs

Bone formation was assessed using high-definition Faxitron radiographs (Figure 4) and micro-CT (Figure 5 and Figure A8). Faxitron assessment showed more moderate/abundant (M/A) bone induction with BMP treatment at L4/5 (*p* ≤ 0.05 for treatment comparisons other than chondroitinase ABC (chABC)/BMP (*p* = 0.102) and BMP/VEGF (*p* = 1)) and at L5/6 (*p* ≤ 0.05 for treatments other than chABC/BMP (*p* = 0.07) and BMP/VEGF (*p* = 1)) (Figure 4). VEGF/BMP had similar behavior, with more faxitron-assessed M/A bone induction at L4/5 (*p* ≤ 0.01) and L5/6 (*p* ≤ 0.001) than any other treatment group, other than BMP (*p* = 1). When Faxitron data were not stratified by level, treatment with BMP or VEGF/BMP induced more M/A bone than any of the other treatments (*p* ≤ 0.001) with the exception of the BMP to BMP/VEGF comparison (*p* = 1). Micro-CT-based bone assessment similarly favored BMP and BMP/VEGF treatment and bone formation (Figure 5), but significance testing was limited to Mock being different from BMP (*p* = 0.004) and BMP/VEGF (*p* = 0.044). Micro-CT also was used to measure bone formed within the disc space, with Regions of Interest designed to not include endplates or bone external to the disc space, but no significant intra-discal bone was detected for any of the treatment groups (*p* ≥ 0.528). We again noted trends for decreased BMP and VEGF/BMP treatment effects (bone production) when combined with chABC. Current data for the combined L4–6 segment demonstrated no differences from our previously published results for faxitron- or micro-CT-based bone formation measurements (Table A2), and induced bone was located anterior to the spine and discs (Figure A8) consistent with our prior findings.

### 3.5. Bone Induction Required BMP Treatment, VEGF Treatment Alone Did Not Induce Angiogenesis, and Chondroitinase ABC Did More Damage Than Expected

Light microscopy outcomes focused on assessing damage induced in the IVD, bone induction internal or external to the disc, and induced angiogenesis (particularly in NP tissue) through use of scoring criteria (Table A1). Composite IVD damage was noted in surgical groups in comparison to Mock (*p* < 0.05 for all but Naïve (*p* = 0.314) and chondroitinase ABC (chABC)/BMP (*p* = 0.082)), and the highest average IVD damage was observed in the chABC and chABC/VEGF treatment groups (Figure 6) but inter-treatment comparisons did not reach statistical significance. Less IVD damage was noted when chABC samples were co-treated with BMP (*p* = 0.036) and when chABC samples were treated with BMP instead of VEGF (*p* = 0.041) (Figure 6 and Figure 7). IVD damage included variable loss of Alcian Blue proteoglycan staining in the NP of chABC compared to Mock, Naïve, VEGF and VEGF/BMP (all *p* < 0.05) and of chABC/VEGF compared to Mock and Naïve (both *p* < 0.034). Operated samples compared to Mock all showed loss of biphasic NP staining (*p* < 0.001), and variable production of collagen stranding in the NP space (demonstrated by positive picrosirus red staining visualized with circular polarized light) that did not demonstrate a treatment effect in post hoc testing. AF waviness and clefts were present more frequently in chABC-treated samples than Mock (*p* = 0.044), and in chABC/VEGF-treated samples than Mock, Naïve, VEGF, BMP and chABC/BMP treatments (*p* < 0.05 for each). Bone induction was most pronounced in the BMP and BMP/VEGF groups (*p* ≤ 0.001 when either was compared with Mock, Naïve, VEGF, chABC and chABC/VEGF), and was also induced in less abundance in the chABC/BMP (*p* < 0.02 compared with Mock, Naïve, VEGF, and chABC/VEGF) and chABC/BMP/VEGF treatment groups (*p* < 0.05 compared with Mock, Naïve, and chABC/VEGF) (Figure 6 and Figure 7), again suggesting antagonism between chABC and the BMP and BMP/VEGF groups, but no significant differences were noted between BMP or BMP/VEGF ± chABC post hoc comparisons. Unexpectedly, angiogenesis into the NP was not significantly detectable in the VEGF monotreatment group, but was observed in the chABC (*p* < 0.02 compared with Mock, Naïve and chABC/BMP treatments) and chABC/VEGF groups (*p* < 0.05 compared with Mock, and *p* = 0.283 when compared with Naïve or chABC/BMP) (Figure 6 and Figure 7). Angiogenesis was frequently associated with a fibrocartilage AF-like discordant healing response where the NP is usually located (cranial-caudal AF-like organized bundles that stain intensely with Picrosirus Red) (Figure 7, see chondroitinase ABC/VEGF). This pattern was most commonly observed in the chABC and chABC/VEGF groups.

### 3.6. Chondroitinase ABC Disrupts Endogenously Expressed VEGF Sequestration, Making the IVD Permissive to Vascular Invasion

We queried the mechanism of Chondroitinase ABC (chABC)-related angiogenesis in our model, and identified a potential role for matrix-bound sFlt. First, we confirmed that chABC causes extensive damage to the NP matrix in our model, including the loss of Alcian Blue staining indicative of decreased proteoglycan content in the NP. Then, recalling that NP cells express VEGF in situ and that the IVD maintains its avascularity despite this VEGF expression must mean that the NP’s VEGF is being sequestered and prevented from diffusing through and external to the disc to recruit vascular ingrowth. We suggest that chABC-mediated matrix proteoglycan damage impairs the sequestration capacity of the NP, releasing the endogenous VEGF to diffuse and induce neo-angiogenesis, suggesting a role for proteoglycans or proteoglycan-associated factors in the sequestration process (Figure 8A). Once sequestration is lost, VEGF-activated vascular tissues respond by invading the IVD and following the new VEGF gradient into the NP compartment where the VEGF was being generated (Figure 8B). In such a model, experimental delivery of VEGF-expressing cells to the NP would potentially have no effect on angiogenesis provided that the sequestration mechanism was sufficiently strong, but co-delivery of chABC and VEGF would be expected to possibly result in an even stronger VEGF diffusion gradient, and stronger angiogenesis responses. This argument then begs the question: what is the mechanism that the IVD uses to sequester the endogenously expressed (or experimentally delivered) VEGF? VEGF sequestration molecules could be expressed throughout the disc, or perhaps only in the NP compartment. Our findings would suggest that candidate sequestration molecules would need to be expressed in at least one of the areas where we observed significant matrix destruction: the NP, CEP and growth plate.

We used two in vitro models to characterize IVD sFlt, a soluble fragment of the VEGF receptor-1 which could bind and sequester VEGF in the IVD matrix. In tissue cultured NP cells from bovine caudal discs (bNPs) we detected sFlt in the cell/matrix portion of the tissue culture samples but not the media (Figure 9). In freshly enucleated rabbit NP tissue from lumbar spines, we observed that treatment with chondroitinase ABC (chABC - causing matrix proteoglycan destruction), but not collagenase type II (matrix protein destruction), solubilizes sFlt from the cell/matrix of fresh rabbit NP tissue (Figure 9). If sFlt functions to sequester VEGF in the NP in vivo, then the chABC-mediated release of sFlt from the NP proteoglycan matrix could mediate the loss of VEGF sequestration and consequent IVD vascularization that we observed. We are continuing to characterize the role that sFlt or other fractional forms of VEGFR1 (sFlt1-14, or possible shed forms generated by ADAMTS enzymes) serve as regulators of NP anti-angiogenesis.

## 4. Discussion

This study has revealed the IVD to be manipulable for angiogenesis, a critical step forward in our evolving rat model to engineer percutaneous IVD fusions for clinical application. We showed that IVD delivery of purified chondroitinase ABC (chABC) resulted in the dramatic disorganization/destruction of the NP compartment, loss of proteoglycan staining in the NP and AF, and angiogenesis but not osteogenesis within the IVD. Adding chABC treatment to delivery of BMP-expressing cells did not result in the predicted bone formation inside the IVD, and the combination resulted in almost no bone formation outside the IVD when compared to BMP-expressing cells alone; however, this chABC/BMP combination also resulted in less chABC-induced IVD damage than chABC alone, suggesting an antagonism between BMP and chABC effects. Codelivery of VEGF and BMP also did not result in bone production inside the IVD, and did not significantly increase the amount of bone induced external to the IVD as compared to BMP alone. Gene delivery of VEGF alone was not sufficient to induce angiogenesis in the IVD, and this surprising result may explain, in part, why the addition of VEGF did not promote bone formation in the IVD in BMP-treated spines, because at a first approximation the absence of VEGF-induced IVD angiogenesis makes the VEGF/BMP condition inside the IVD work like BMP alone. Triple treatment with chABC/BMP/VEGF was most similar to VEGF or chABC/VEGF, with some damage and angiogenesis observed but no IVD bone formation, either showing another example of chABC-BMP antagonism or suggesting that factors other than the absence of vascularization inhibit bone formation within the IVD. Finally, we confirmed that the BMP gene delivery technique is reproducible for increasing spinal stiffness, induced bone formation and segmental fusion when the current data were compared with our prior findings from identically treated animals.

A recent Scoping Review examined the English language literature for human disc vascularization, and found no evidence for NP vascularity throughout life, while CEP and AF have early vascularity that involutes at maturity and returns again during degeneration and aging [27]. The non-degenerate NP at maturity is dependent on diffusion for delivery of nutrients and removal of waste [57], is considered to be the largest avascular organ [27,57], and expresses the hypoxia-inducible factors that would be required to up-regulate angiogenesis signaling molecules capable of inducing vascular invasion into the disc space [57,58]. However, despite multiple reports showing that NP cells express VEGF [29,30,31,32], the NP remains avascular, as if the NP-expressed VEGF was being sequestered and “hidden” within the IVD to avoid being detected and acted upon by VEGF-sensitive vascular progenitors that could drive angiogenesis. The avascular cornea uses such a system, where the VEGF-expressing corneal cells co-express s-Flt (a soluble form of the VEGF receptor-1) that functions to sequester VEGF from the blood vessels of the sclera, and when corneal s-Flt expression is experimentally attenuated, VEGF becomes detectable and aggressive angiogenesis onto the cornea is induced [59]. The IVD has been reported to express s-Flt [31,60], and this sequestration mechanism for IVD anti-angiogenesis has been suggested, but has not previously been demonstrated [60]. IVD avascularity is very well preserved in vertebrates [31,61], suggesting that a robust set of anti-angiogenesis mechanisms maintains homeostasis, as is also the case for the cornea (including thrombospondin-1 (TSP1) [62], TSP2 [63], TSP3 [64], tissue inhibitor of metalloproteinases-3 (TIMP3) [65], TIMP4 [66], among others [67,68]). Sequestration/regulation of VEGF by s-Flt may be a generalizable mechanism, having also been described for several tissues, including ovary [69], kidney [70], placenta [71], hemogenic/endothelial progenitors [72], and others [73,74].

We observed that delivery of chondroitinase ABC (chABC) to the IVD rendered the disc permissive to vascular invasion, and that chABC co-administered with VEGF gene delivery also drives IVD angiogenesis. We have presented preliminary in vitro evidence suggesting that extracellular matrix (ECM)-associated sFlt is deregulated by chABC matrix proteoglycan destruction, likely through the solubilization of sFlt from its attachment to proteoglycans in the ECM, thereby rendering it incapable of sequestering VEGF to maintain IVD avascularity. The sFlt was not solubilized by collagenase type II, suggesting that ECM protein destruction is not sufficient for IVD angiogenesis. NP angiogenesis was not observed in the absence of chABC, regardless of delivery of Naïve cells, VEGF-expressing, or VEGF-and-BMP-expressing cells, demonstrating that the endplate perforations and surgical insult is not sufficient for IVD angiogenesis. During disc degeneration, similar damage to the IVD occurs, including loss of NP ECM, but vessel ingrowth into the NP is not robust [27], suggesting that still other anti-angiogenesis mechanisms are present. The NP is a notochord-derived tissue, and as such expresses multiple patterning genes that would be capable of attracting or repelling specific cell populations to establish and maintain the very specific environment of the NP interior [32,35], including repelling vascular invasion (shown in non-IVD tissues) through WISP-2/CCN5 [75], Semaphorin 3E [76] and 3A/Collapsin-1 [77], Noggin [78,79], Endothelin-2 [80], Chordin [78,79], and notochordal chondroitin sulfate [77]. If there was direct evidence that they provided anti-angiogenesis effects in the IVD, any of these notochordal angiogenesis-controlling molecules would fit our data, if it can be assumed that their effect decreases as the IVD cell population vitality does, and that the NP is basically destroyed following chABC treatment. For example, the semaphorins are a set of morphogens that control both nerve and blood vessel patterning, and have been suggested to be a mechanism maintaining IVD avascularity through endothelial cell repulsion [81,82]. Semaphorin 3A is a diffusible factor that is expressed in the IVD at higher levels in the AF than NP, but during degeneration AF expression drops while NP expression increases [20,81,82]. Semaphorin 3A expression would appear to correlate with what is observed with degeneration and AF vascular invasion, and would appear to correlate with what has been reported about the NP remaining relatively avascular despite degeneration [27], but this candidate needs direct assessment as regards its expression levels and function in the setting of chABC treatment.

We observed a correspondence between NP compartment destruction, induced angiogenesis in that location, and an atypical inner-AF fibrocartilage healing response in the IVD that extended through the disc and into the juxtaposed vertebral bodies. This is similar to the appearance of IVDs where HIF-1alpha [36,38] or SHH [37] knockdown resulted in the loss of notochordal NP cells, causing shrinking and degeneration of the NP compartment, and NP replacement by fibrocartilaginous tissue resembling the inner-AF [36,38]. Angiogenesis and the aberrant fibrocartilage healing response may both be examples of IVD homeostasis disruption, secondary to the loss of notochord-dependent inhibitory signaling in the setting of the destroyed NP, that results in the release of both anti-angiogenesis and anti-proliferation of the inner-AF tissues. Further characterization of this fibrocartilage tissue and its behavior is warranted.

The antagonism between BMP and chondroitinase ABC (chABC) treatments is thought to be driven by their differential effects on matrix proteoglycans. The lack of BMP-driven osteogenesis external to the IVD in chABC-treated spines is likely due to impaired endochondral ossification. We previously noted Alcian blue-stained cells between the anterior longitudinal ligament and anterior fibers of the AF in non-fused samples in which fusion bone was otherwise observed [7], suggesting the surgical trauma produced a progenitor population of cells capable of responding to a sufficiently strong osteoinductive signal (such as BMP-expressing cells) by undergoing endochondral osteogenesis. This BMP-driven process would include the production of an extracellular matrix rich in chondroitin/dermatan sulfate-containing (CS/DS) proteoglycans (aggrecan, versican, decorin, and biglycan), that would eventually undergo conversion to bone [83]. As chABC breaks down CS/DS proteoglycan-rich matrices, impaired endochondral osteogenesis and lack of bone formation may not be surprising [84,85]. Within the IVD, on the other hand, the chondrogenic activity of BMP likely limited the chABC-driven IVD destruction. This finding is consistent with other studies, in which BMP treatment drove IVD PG synthesis sufficient to restore radiographic disc height, biochemical assessments, and T2-weighted MRI hydration signal [86,87]. As to which activity dominates (chABC vs. BMP), timing and duration of action likely play critical roles. Adenoviral vectors with CMV promotors classically have about a week of high transgene expression, and then waning but detectable transgene expression out to 2–4 weeks [25,88,89,90]; chABC is very temperature-sensitive and has a half-life of about 12 h at physiological temperatures [91], although one-time mega-dosing has been reported to allow effectiveness for as long as 3 weeks [92]. Future work could include varying the dose of chABC delivered in our model, to decrease its duration of effect and potentially allow it to act towards angiogenesis, but block the anti-osteogenesis effect.

We observed that VEGF did not augment BMP-driven osteogenesis in our model, in contrast to previously published results, which have reported that their combined application increased bone formation, and that the mechanism involves each gene upregulating the other [39,42,43]. One reason for the difference between our observation and the prior reports may be our use of the hyper-osteoinductive BMP2/BMP7 heterodimer form of these BMPs [93,94]. In our model the VEGF-treated specimens may have generated expression of homodimer forms of BMPs, but these homodimer BMPs were not sufficiently osteoinductive to drive bone formation and fusion in the model, as we have previously shown [7]. Alternate possibilities are that negative feedback mechanisms driven by the heterodimer osteoinductive signaling suppressed VEGF-mediated endogenous BMP upregulation, or that optimal VEGF:heterodimer-BMP ratio is not 1:5, as has been shown for homodimer BMPs, or that the study was underpowered to detect the augmentative effect. Regardless of the molecular mechanisms, an interesting observation from our model is that angiogenesis occurring in chondroitinase ABC (chABC)/VEGF cotreated discs was not sufficient to support BMP-mediated osteogenesis in the IVD in the chABC/VEGF/BMP treatment condition. Therefore, the presence of a proteoglycan-rich matrix and lack of blood supply may not be the only factors preventing bone formation in the IVD milieu. Future work to further investigate these possibilities is needed to develop a percutaneous method of intradiscal spinal fusion.

## 5. Limitations

The current work has several limitations. Our study was designed to use NIAD to assess induced spinal stiffness over the 12 weeks of the experiment, which did not require euthanasia nor did it allow histology or other assessments at intermediate time points. We were not able to track the implanted cells, monitor implanted cell viability, or measure transgene expression levels from implanted cells after implantation. Our chondroitinase ABC (chABC) dose was chosen by a critical assessment of the published literature, but as judged by near-complete loss of BMP-induced bone in the chABC/BMP and chABC/BMP/VEGF codelivery conditions, it may have been too high. Future experiments to optimize chABC are warranted to investigate if a lower chABC dose could help support osteoinduction as originally hypothesized. Similarly, we did not identify the optimal ratio of VEGF:heterodimer-BMP for VEGF/BMP codelivery, and a different ratio may have improved osteogenesis as we expected. We chose the VEGF_165_ subtype by comparing the known properties for how well the isoforms diffused through the matrix, where the largest (VEGF_206_) did not diffuse well and may not be detected external to the IVD, the smallest (VEGF_121_) diffused very well and may not allow a concentration gradient to be established to direct the angiogenesis into the IVD, making the predominant mid-sized subtype (VEGF_165_) our favored choice [95]; future work could experimentally assess these other VEGF subtypes to determine the optimal size for this surgical model. We have suggested that sFlt may play a role in sequestering IVD-expressed VEGF and IVD anti-angiogenesis homeostasis, but further investigation is required to validate this mechanism. Similarly, the potential role (or presence) of H-type capillaries (the principal endothelial cell type associated with bone, identifiable by their high expression of endomucin and CD31) [96] was not assessed, and needs to be evaluated in future use of the model. Lastly, the fibrocartilage proliferation/healing response in the IVD requires further study to characterize and understand what is being observed.

## 6. Conclusions

The delivery of chondroitinase ABC (chABC) to the IVD in our rodent model made the NP compartment of the disc permissive to angiogenesis, but chABC co-treatment nearly abolished BMP-induced bone formation external to the disc space, abolishing fusion ability in those samples. Further work is necessary to modify the current, or engineer an equally effective alternate, angiogenesis treatment that enables osteogenesis, or develop an osteogenesis treatment resistant to the effects of chABC. We plan to continue investigating the molecular mechanisms controlling NP anti-angiogenesis, alternate ways to affect the NP extracellular matrix or NP expressome, the use of allogeneic implanted cells, and larger comparative models.

Using intradiscal gene- and/or factor-delivery, we have demonstrated that our method for driving bone formation and spine fusion is reproducible, and that angiogenesis can be induced in the NP compartment of the IVD. These observations are definite steps towards our goal to develop a method to induce percutaneous human anterior spine fusions, a technique that would reduce pain and suffering, and reduce or eliminate hospital stays and lost days at work.

## Figures and Tables

**Figure 1 cells-12-02488-f001:**
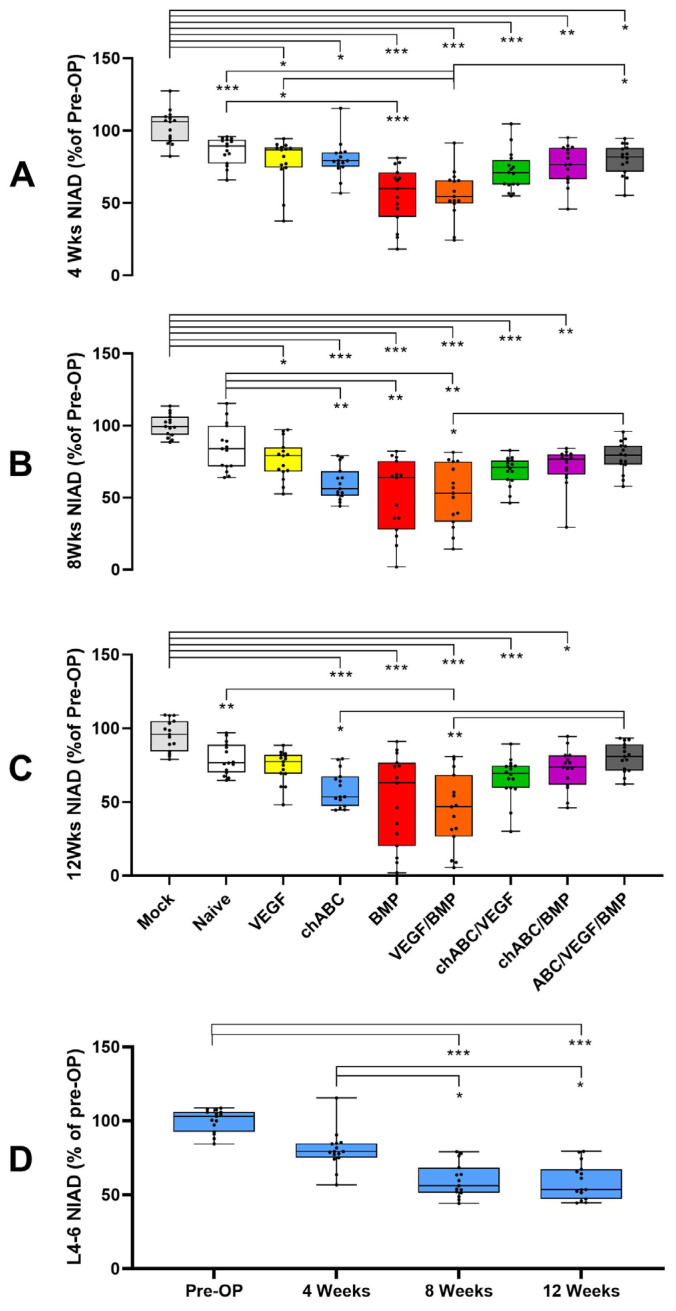
Coronal NIAD assessments for the combined L4–6 motion segment after L4/5 and L5/6 intra-discal delivery of indicated treatments. Panels represent all treatment groups at 4 weeks (**A**), 8 weeks (**B**) and 12 weeks (**C**) post-operatively, or the chondroitinase ABC (chABC) treatment group at 0, 4, 8 and 12 weeks (**D**). Indicated significance: *p* ≤ 0.001 (***), *p* < 0.01 (**), and *p* ≤ 0.05 (*).

**Figure 2 cells-12-02488-f002:**
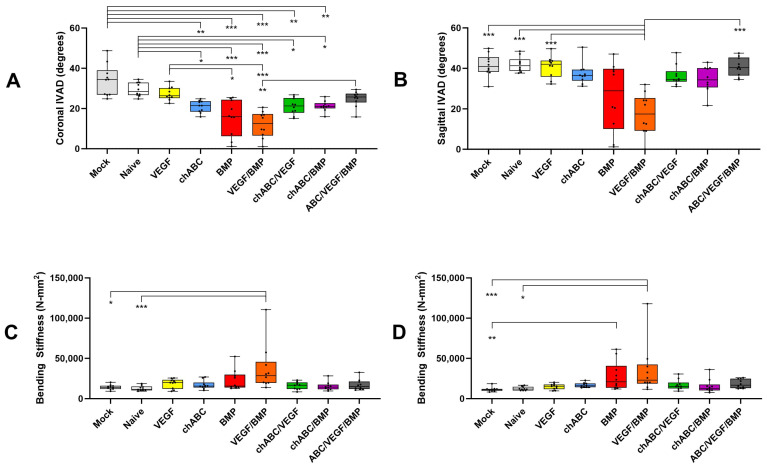
In vitro angular displacement (IVAD) and non-destructive four-point bending mechanical assessments of the combined L4–6 motion segment at 12 weeks post-implantation of the indicated treatments. IVAD is shown in in the coronal (**A**) and sagittal (**B**) planes, and four-point bending stiffness is shown in the left (**C**) and right (**D**) bending directions. Chondroitinase ABC is abbreviated as chABC. Indicated significance was *p* ≤ 0.001 (***), *p* < 0.01 (**), and *p* ≤ 0.05 (*).

**Figure 3 cells-12-02488-f003:**
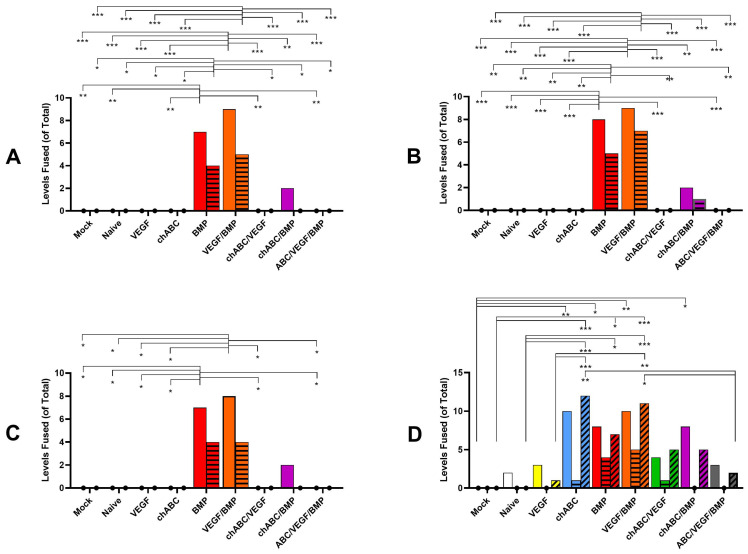
Multimodal spinal fusion assessments at 12 weeks post-implantation of the indicated treatments. Fusion status was scored separately at the L4/5 (histograms with no lines) and L5/6 (histograms with horizontal lines) levels for all assessments, and for the combined L4–6 (histograms with diagonal lines) segment for the NIAD fusion prediction method. Fusion was assessed by palpation (**A**), categorical radiographic (**B**), graded radiographic (**C**), and critical NIAD values (**D**), as described in the Materials and Methods section. Histogram vertical heights indicate raw value for number of fusions, and group size was *n* = 15 for all treatments except chondroitinase ABC (chABC)/VEGF (*n* = 16). Indicated significance was *p* ≤ 0.001 (***), *p* < 0.01 (**), and *p* ≤ 0.05 (*).

**Figure 4 cells-12-02488-f004:**
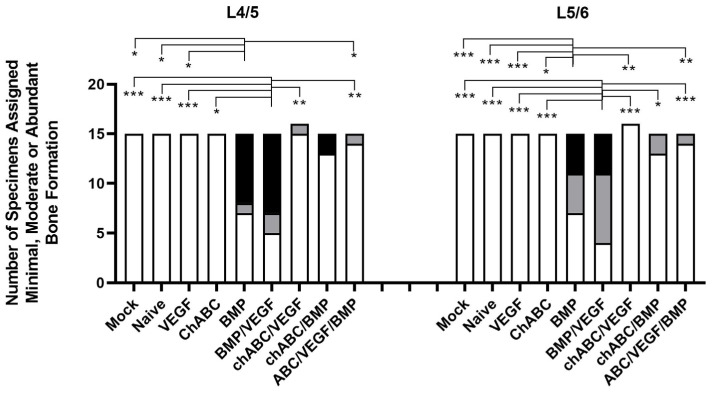
BMP-driven bone induction as assessed by high-definition Faxitron radiographs. Groups are composed of *n* = 15 or 16 specimens, the L4/5 and L5/6 levels were scored independently, and scoring and data testing were conducted as described in M&M. Histogram filled with white background indicates Minimal bone formation, gray indicates Moderate bone formation, and black Abundant bone formation. Chondroitinase ABC is abbreviated as chABC. Indicated significance was *p* ≤ 0.001 (***), *p* < 0.01 (**), and *p* ≤ 0.05 (*).

**Figure 5 cells-12-02488-f005:**
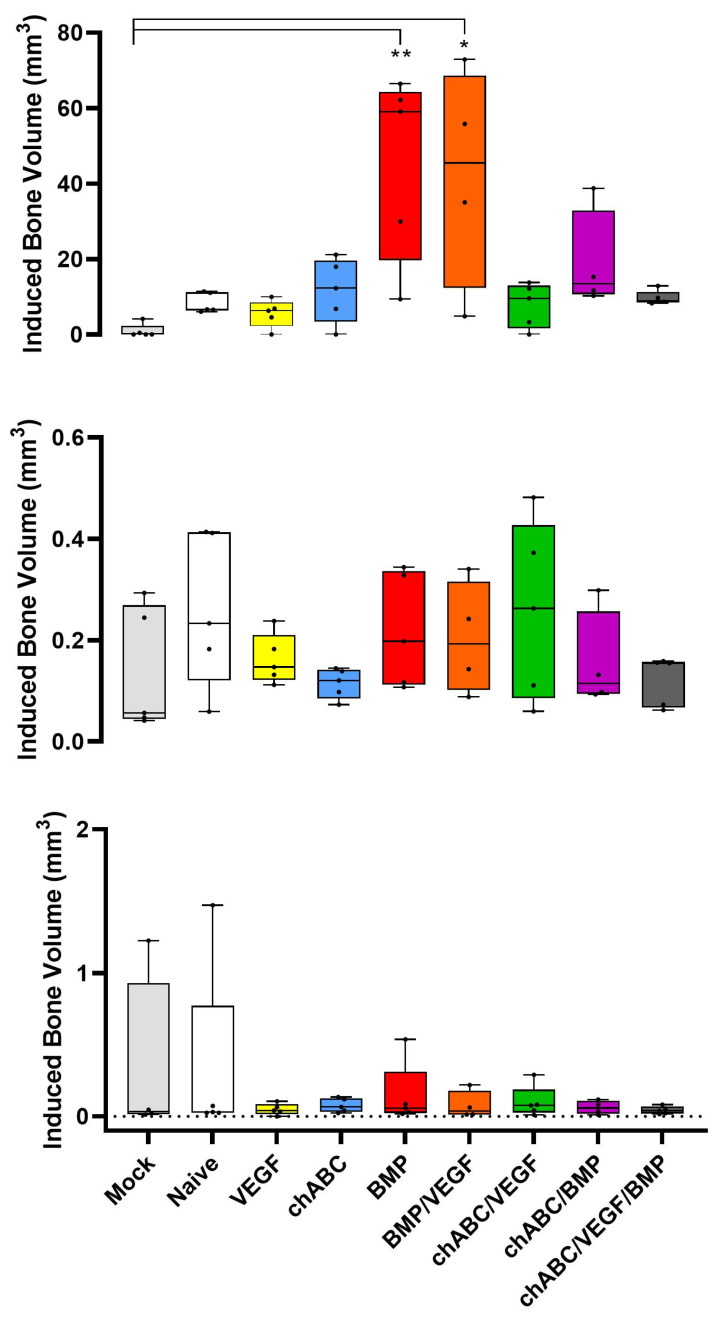
BMP-driven bone induction as assessed by micro-CT. Groups are composed of *n* = 4 (BMP/VEGF and chondroitinase ABC (chABC)/BMP) or *n* = 5 (all other groups) specimens. Specimens were prepared, assessed, and data tested as described in M&M. Top panel shows quantification of induced bone for the entire L4–6 fusion mass (located anterior to the spine and discs), the middle panel shows bone formation inside the L4/5 disc, and the bottom panel shows bone formation inside the L5/6 disc. Indicated significance was *p* < 0.01 (**), and *p* ≤ 0.05 (*).

**Figure 6 cells-12-02488-f006:**
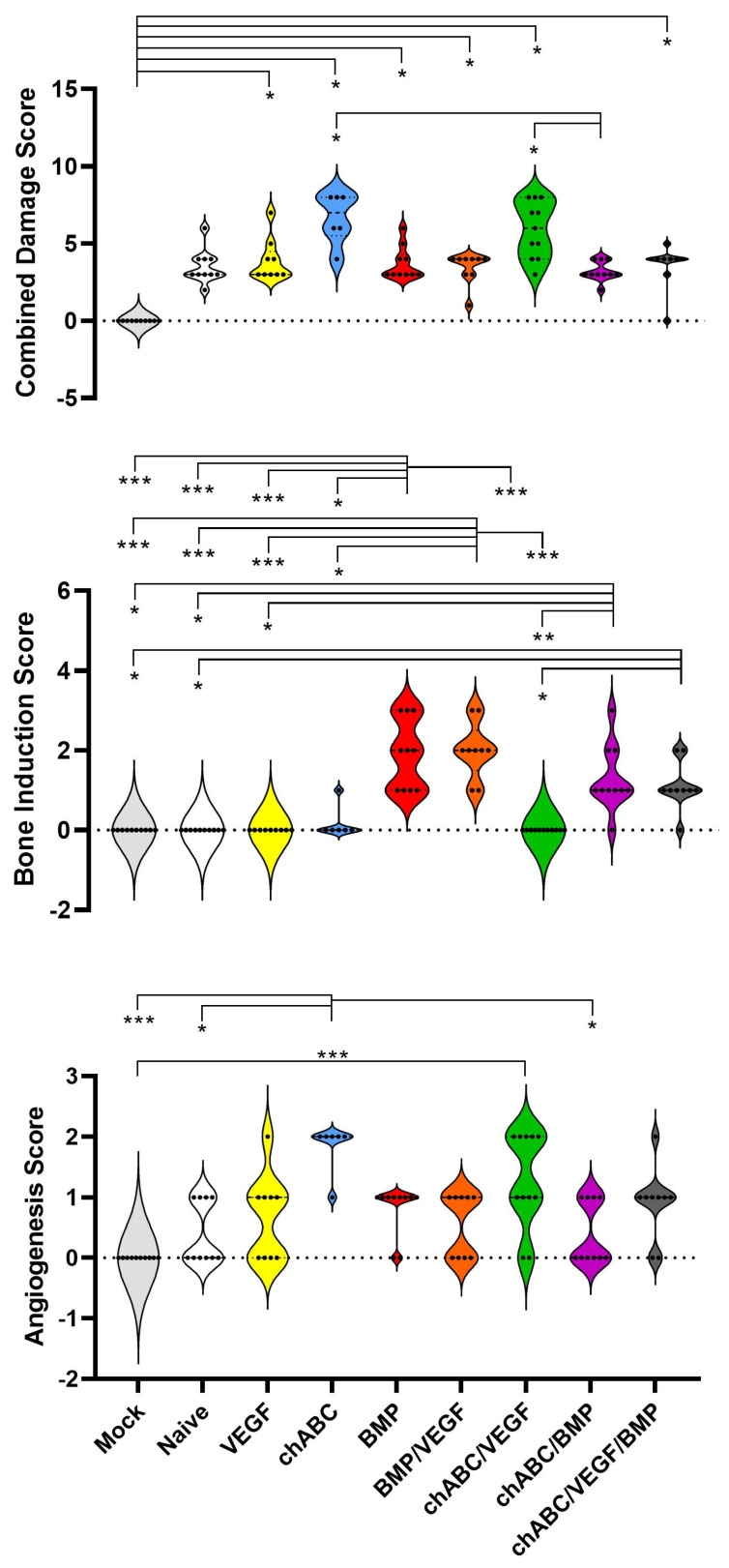
Disc histology scoring results at 12 weeks after intradiscal delivery of treatments. Using a scoring rubric (Table A1), specimens were assessed for IVD damage criteria (top panel, range 0 to 8), osteogenesis extent (middle panel, range 0 to 3) and evidence of angiogenesis (bottom panel, range 0 to 2), as described in the Materials and Methods section. Results are presented using violin plots to demonstrate the distribution of the ordinal data. Chondroitinase ABC is abbreviated as chABC. Indicated significance was *p* ≤ 0.001 (***), *p* < 0.01 (**), and *p* ≤ 0.05 (*).

**Figure 7 cells-12-02488-f007:**
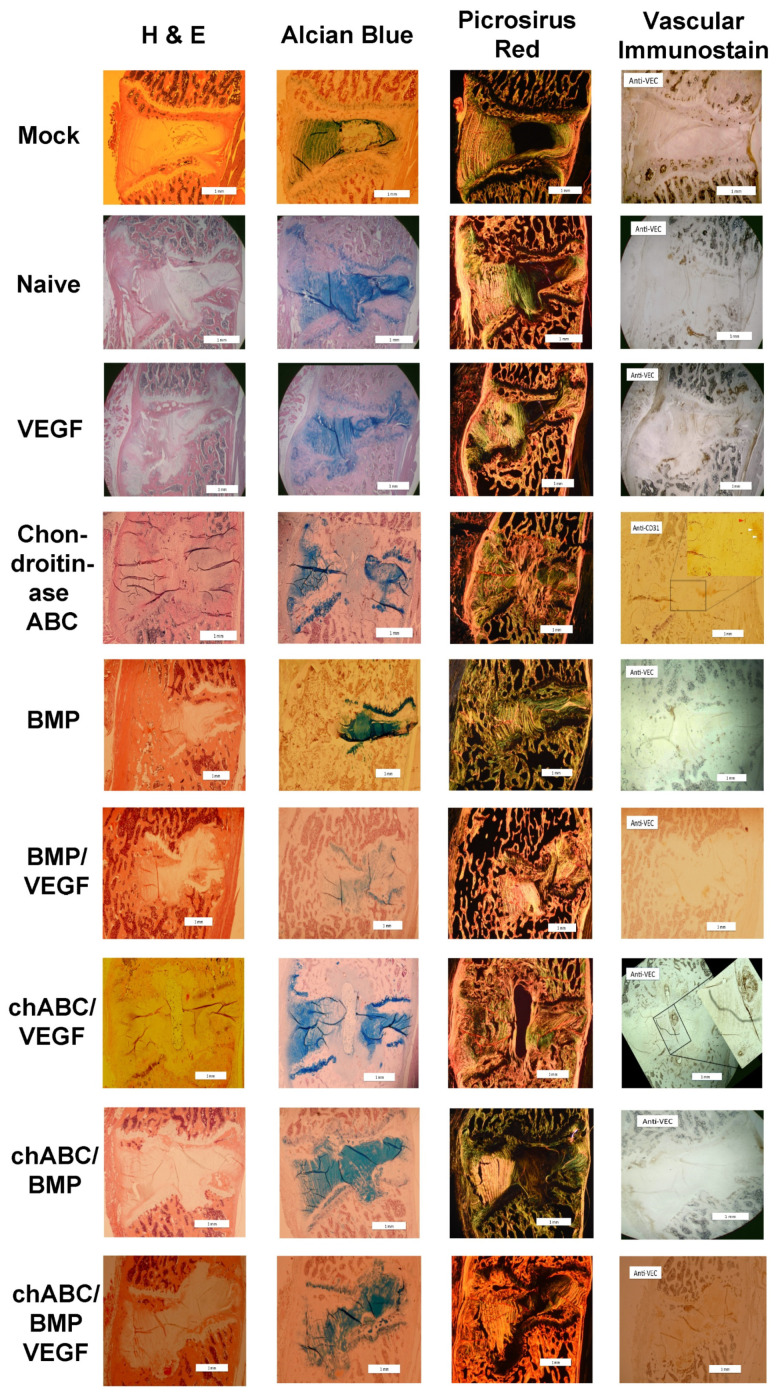
Representative histology images at 12 weeks after intradiscal delivery of treatments. All images are oriented ventral to the left and cranial as top. Histological staining technique is indicated at the top of each column and treatment group is indicated on the far left of each row. Picrosirus Red birefringence was visualized with circular polarized light, and vascular immunostain shown in the figure is anti-VE-cadherin or anti-CD31 (+), as indicated in each panel in that column (upper-left corner). Arrowheads in immunostained images indicate vascular appearing (red arrowhead) or cell clusters (white arrowheads) with positive immunostain. Scale bars in the right lower corner indicate 1 mm. Chondroitinase ABC is abbreviated as chABC.

**Figure 8 cells-12-02488-f008:**
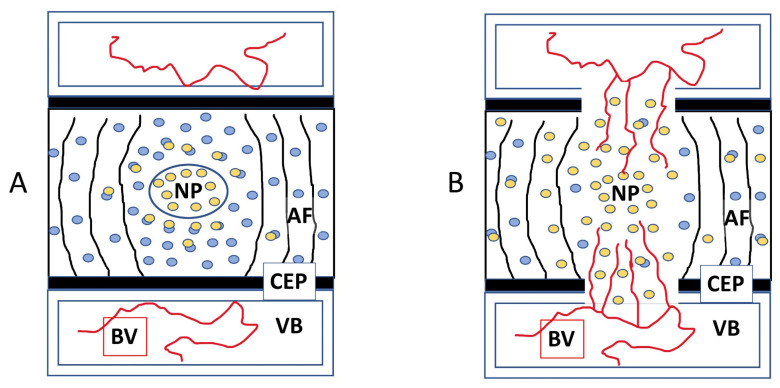
A model for sFlt deregulation in the NP and how it would lead to vascular invasion of the IVD/NP. Major involved structures are labeled with initials: NP = nucleus pulposus, AF = annulus fibrosus, CEP = cartilage endplate, VB = vertebral body, and BV = blood vessels in proximity. Prior to endplate injury and sFlt deregulation, VEGF (yellow ovals) is being bound and sequestered from detection by tissues external to the disc space by sFlt (blue ovals) as shown (**A**). After sFlt is deregulated, there is reduced sFlt presence in the NP, VEGF is able to diffuse further in all directions, and particularly towards the endplate perforations where VEGF-sensitive blood vessels are positioned and are able to contribute to angiogenesis into the IVD as shown (**B**).

**Figure 9 cells-12-02488-f009:**
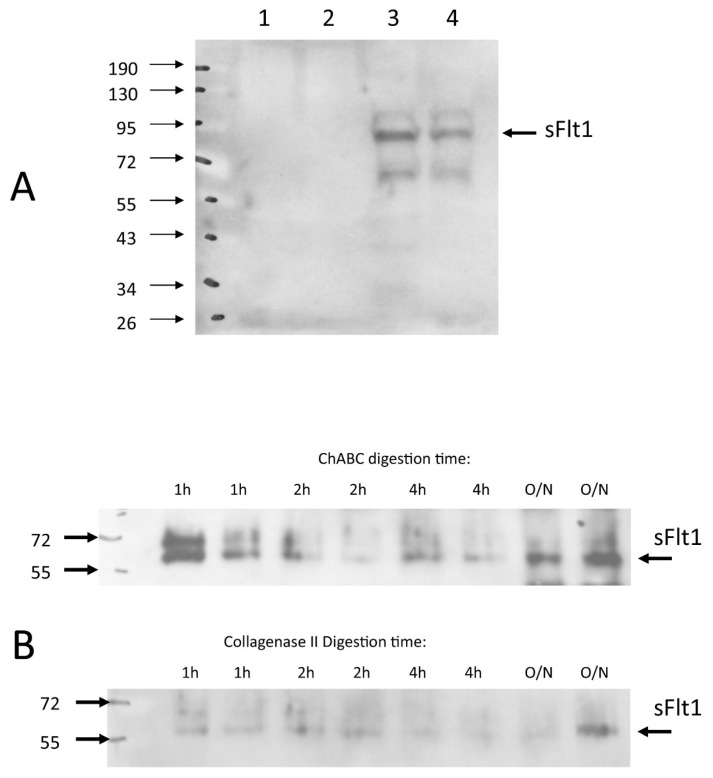
sFlt Western Blotting of bovine NP cells and rabbit NP tissue. Western immunoblotting for soluble VEGF-R1 (sFlt) demonstrated that sFlt was not detectable in supernatant media in bovine NP monolayers ((**A**), lanes 1 and 2), but rather was found in the cell/matrix fraction ((**A**), lanes 3 and 4) and this was not altered by normoxia (21% oxygen culture conditions, lanes 1 and 3) or hypoxia status (2% oxygen culture conditions, lanes 2 and 4). Freshly enucleated rabbit NP tissue when subjected to chondroitinace ABC (ChABC in the figure) digestion over time course (see Materials and Methods) released sFlt from the cell/matrix fraction ((**B**), upper blot), but digestion of sister samples with Collagenase Type II did not consistently solubilize sFlt ((**B**), lower blot).

## Data Availability

Data generated or analyzed during this study are available from the corresponding author on reasonable request.

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
