# Peer review of "Inducing Angiogenesis in the Nucleus Pulposus"

_cells, 2023, doi:10.3390/cells12202488_

Round 1

Reviewer 1 Report

This study investigated bone morphogenetic protein (BMP) gene delivery to rat intervertebral discs (IVDs), which typically resist bone formation. The researchers explored methods to make IVDs more conducive to BMP-driven bone formation and fusion. They used proteoglycanase to destroy IVDs and an angiogenic growth factor to enhance blood supply, alongside gene-programmed cells expressing BMP2/BMP7 or VEGF165. These treatments were applied surgically between lumbar vertebrae. Results showed that BMP and VEGF/BMP treatments yielded the best outcomes in terms of stiffness, bone production, and fusion. Interestingly, bone formed outside the IVD, with chondroitinase-ABC impairing BMP-driven bone formation but permitting angiogenesis. The study suggested that manipulating the IVD could encourage vascular invasion and potentially osteogenesis.

This is a very interesting study with robust data and novel findings.

Please conduct an analysis and incorporate data regarding type H vessels, utilizing methods such as immunostaining or RNA expression analysis of CD31 and Endomucin,  PMID: 34281770.

Ensure that scale bars are added to all the images present in Figure 4.

Kindly provide information regarding the age and gender of the rats utilized in the study, as these details are currently missing.

Additionally, it would be valuable to ascertain whether the experiments were conducted on aged rats?

Minor editing

Author Response

Reviewer #1

Summary –

This study investigated bone morphogenetic protein (BMP) gene delivery to rat intervertebral discs (IVDs), which typically resist bone formation. The researchers explored methods to make IVDs more conducive to BMP-driven bone formation and fusion. They used proteoglycanase to destroy IVDs and an angiogenic growth factor to enhance blood supply, alongside gene-programmed cells expressing BMP2/BMP7 or VEGF165. These treatments were applied surgically between lumbar vertebrae. Results showed that BMP and VEGF/BMP treatments yielded the best outcomes in terms of stiffness, bone production, and fusion. Interestingly, bone formed outside the IVD, with chondroitinase-ABC impairing BMP-driven bone formation but permitting angiogenesis. The study suggested that manipulating the IVD could encourage vascular invasion and potentially osteogenesis.

This is a very interesting study with robust data and novel findings.

Points –

  1. Please conduct an analysis and incorporate data regarding type H vessels, utilizing methods such as immunostaining or RNA expression analysis of CD31 and Endomucin,  PMID: 34281770.

This is a fantastic idea! It is not feasible for us to perform these experiments for the current manuscript, but it is a very insightful suggestion, and we will definitely include this in our upcoming grant submission!

The concept was included in the latter half of the ‘Limitations’ paragraph at the end of the Discussion, it is in yellow highlight text.

“Similarly, the potential role (or presence) of H-type capillaries (the principal endothelial cell type associated with bone, identifiable by their high expression of endomucin and CD31)[97] was not assessed, and needs to be evaluated in future use of the model.”

  1. Ensure that scale bars are added to all the images present in Figure 4.

Excellent point, I overlooked this while prepping the images for the figure.  These have been added.

  1. Kindly provide information regarding the age and gender of the rats utilized in the study, as these details are currently missing.

The rats were all male (to avoid issues regarding gender in this early development of the model) and were all ~200-250 grams at time of surgery (about 8 weeks), but future experiments will definitely include male and female animals, as well animals of varying ages.  

This information was added into the M&M to better describe the animal cohort.  This is in section 2.1, first and third paragraphs, and is in yellow highlight.

  1. Additionally, it would be valuable to ascertain whether the experiments were conducted on aged rats?

Here we know that the animals were of a specific age/weight and were not aged.  But otherwise, yes!, it would be interesting to see if the model is successful (as we describe here) in females and in animals of varying ages.  More things to consider in the future application of the model.

Reviewer 2 Report

Sheela et al. performed intervertebral disc injection of cells that expressed BMP2/7, VEGF, chondroitinase-ABC (chABC), or combination of these reagents in Lewis rats. The authors stated that treatment with BMP and VEGF/BMP had the highest stiffness, bone production and fusion. Treatment with chABC impaired BMP-driven osteogenesis, decreased proteoglycan staining, and mage the disc permissive to angiogenesis. I have some suggestions to the authors.

1.     Introduction: It is unclear why patients need segmental spinal fusion surgery. The author should describe what kind of patients require this surgery, and what kinds of benefit do spinal fusion provide to these patients.

2.     Results:

a.     The Materials and Methods described that “Spine were prepared for micro-CT” (Line 180). However, no micro-CT images were displayed in the results section. The authors should display micro-CT images for all the groups.

b.     The authors described that BMP and BMP/VEGF decreased in vivo angular displacement (NIAD). It is unclear if reduction of NIAD is good or bad for patients? The authors should explain this in the introduction section, such as some of the parameters related to spinal disease.

c.     The authors described that BMP and VEGF/BMP increase stiffness. It is unclear if the increase of stiffness benefit to patients? The authors should explain it in the introduction section.

d.     In figure 4, the authors showed some histological images of tissue sections.  First, it is better to list the staining on the top of images. It is better to list study groups on the left of images. Currently, the authors only described that A=mock, B=Triple, C=BMP. It is unclear what treatment in D and E. The author should display all the treatment groups in the figures. The background was different among all the tissue sections. It is better to adjust background to similar level. It is unclear what kind of immunostaining for vascular immunostain in the figures. The material sections mentioned for cadherin immunostain and CD31 immunostain. At the top of images, it is better to list all the staining methods.

e.     Because chABC dissolve disc tissues, it might cause damage to disc tissues. Was the angiogenesis response induced by chABC associated with tissue damage induced by chABC?

3.     Discussion

It is unclear if intervertebral disc injection of adenovirus-encoding BMP or BMP/VEGF is safe for patients. It is unclear what are the benefit of this intervertebral disc injection of adenovirus-encoding BMP or BMP/VEGF to patients.

Author Response

Reviewer #2

Summary –

Sheela et al. performed intervertebral disc injection of cells that expressed BMP2/7, VEGF, chondroitinase-ABC (chABC), or combination of these reagents in Lewis rats. The authors stated that treatment with BMP and VEGF/BMP had the highest stiffness, bone production and fusion. Treatment with chABC impaired BMP-driven osteogenesis, decreased proteoglycan staining, and mage the disc permissive to angiogenesis. I have some suggestions to the authors.

Points –

  1. Introduction: It is unclear why patients need segmental spinal fusion surgery. The author should describe what kind of patients require this surgery, and what kinds of benefit do spinal fusion provide to these patients.

An explanation was added at the beginning of the first paragraph of the Introduction (yellow highlighted section).

“Patients experiencing intolerable axial spine pain refractory to non-operative treatment, either in the setting of spinal instability (hypermobility due to fracture, spondylolisthesis, or end-stage spondylosis), regional spinal deformity (scoliosis, kyphosis), or certain spinal infections/neoplastic diagnoses, are frequently indicated for segmental spinal fusions as definitive treatment. Fusion of the affected spinal segments limits or eliminates axial pain by stabilizing treated hypermobile segments, correcting current and preventing progressive spinal deformity, and augmenting structural integrity of the spine in situations of impending pathological fractures.”

  1. Results:
  2. The Materials and Methods described that “Spine were prepared for micro-CT” (Line 180). However, no micro-CT images were displayed in the results section. The authors should display micro-CT images for all the groups.

Isosurface rendered 3-dimensional micro-CT images were included as a new figure in the Supplementary section.  This is Figure A7, and has a representative example of one spine from each experimental treatment group.

  1. The authors described that BMP and BMP/VEGF decreased in vivo angular displacement (NIAD). It is unclear if reduction of NIAD is good or bad for patients? The authors should explain this in the introduction section, such as some of the parameters related to spinal disease.

A discussion of fusion used clinically was added to the first paragraph (immediately following the addition noted above in Point #1), and is in Green highlighted text.  A better description of NIAD was placed in Introduction paragraph #2 (and is in Yellow highlighted text), an attempt is made to parallel concepts from clinical medicine, to make it clear how NIAD is useful. 

Paragraph #1 text added:

“Current surgical technique includes using metal implants to immobilize the spinal vertebrae to be fused, decortication of strategic surfaces of the vertebrae to induce local osteogenesis mechanisms, and placement of bone grafts to optimize osteoconduction. The expectation is for an initial non-mineralized osteoid anlage connected to the spinal vertebrae to be generated, for the initial osteoid to undergo mineralization over time transforming it into a mature fusion mass, and that as the fusion mass evolves/mineralizes, for the spine to progressively stiffen until no motion is detectable, resulting in spinal fusion.  “

Paragraph #2 text added:

“We used non-invasive induced angular displacement (NIAD) at 4-week intervals to measure the loss of spinal motion over time, and showed the pre-operative NIAD measurement decreased in all surgical groups (attributed to soft tissue scarring) by ~25%, but only the heterodimer group showed progressive NIAD losses beyond 4 weeks. BMP-induced bone formation was dose-responsive to the relative osteoinductiveness of the treatment delivered (heterodimer (BMP2/BMP7) > homodimers (BMP2 or BMP7 or mixed BMP2 + BMP7) > negative control (betagalactosidase)), but significant spinal fusion was only observed for the heterodimer treated group. Progressive loss of NIAD over the course of the experiment predicted relative bone production, increased stiffness assessed by 4-point bend biomechanics, and spinal fusion (assessed by palpation and radiographic endpoints). “

  1. The authors described that BMP and VEGF/BMP increase stiffness. It is unclear if the increase of stiffness benefit to patients? The authors should explain it in the introduction section.

This is addressed by the text additions that we made in Introduction paragraph #1, in Point #1 and 2.b, above.  It was explained that fusions are our current method for treating patients in pain from several diagnoses, and that stiffness is an evolutionary stage of fusion, where the spine goes from mobile/painful to stiff to fused/comfortable.

  1. In figure 4, the authors showed some histological images of tissue sections.  First, it is better to list the staining on the top of images. It is better to list study groups on the left of images. Currently, the authors only described that A=mock, B=Triple, C=BMP. It is unclear what treatment in D and E. The author should display all the treatment groups in the figures. The background was different among all the tissue sections. It is better to adjust background to similar level. It is unclear what kind of immunostaining for vascular immunostain in the figures. The material sections mentioned for cadherin immunostain and CD31 immunostain. At the top of images, it is better to list all the staining methods.

Excellent points, thank you for pointing these out.  Labels describing the staining used were added to the figure image along the top, and labels listing the treatment groups were added to the image along the left border.  Images from all treatment groups were included, and I adjusted the backgrounds to be as equal as possible, without altering the ability to interpret the images (without distorting the actual color too much).  Labels for what particular vascular immunostain used for each panel were placed in the upper left corner of the images.  We also added scale bars to every image (as suggested by another reviewer).

  1. Because chABC dissolve disc tissues, it might cause damage to disc tissues. Was the angiogenesis response induced by chABC associated with tissue damage induced by chABC?

We believe that what you say is correct, that the chABC damage is what allowed the angiogenesis response to happen.  The Discussion was re-ordered and re-worded to try to make this concept more clear.  It is explained that the groups with the most damage (chABC and chABC/VEGF) had the most robust angiogenesis, and those other groups with less damage had less angiogenesis.  Unfortunately, it is a bit of a correlative/circumstantial argument at this point – we need to get a bit more findings regarding molecular mechanism, and we will be able to make much more definitive statements about this Hypothesis.  Thank you for the question!

  1. Discussion

It is unclear if intervertebral disc injection of adenovirus-encoding BMP or BMP/VEGF is safe for patients. It is unclear what are the benefit of this intervertebral disc injection of adenovirus-encoding BMP or BMP/VEGF to patients.

Safety for the delivery of the intra-discal treatment is basically asking the question: is it safe to deliver purified chABC and/or Adenovirus-vectors containing genes of interest.  For the former, we discussed in the introduction the safety profile of chABC delivered to several tissues, including the IVD, in comparative surgical models.  It is a compound currently being used to study spinal cord injury healing/re-growth, so, in addition to the literature we cited, there are additional reports suggesting it is safe when applied to the central nervous system directly.  As regards Adenovirus (Ad), we also showed the use of Ad in clinical trials for delivery of genes of interest to heart, liver and pulmonary tissues.  The disc would be expected to be as safe, or possibly safer, as there is a very poor blood supply in the disc, and the virus delivered there would much more likely be contained in the disc than in other tissues such as lung, heart or liver that have robust blood supplies.  Unfortunately, only a true clinical trial will get us to a “real answer”.

As to why someone would want their disc injected with Ad-BMP or Ad-BMP/VEGF, and what they stood to benefit by doing so, the answer lies in our initial goal of this line of investigation: spinal fusion, with minimal risk.  Our goal has been to generate a treatment that could be injected into the disc space and turn the disk into bone, resulting in a spinal fusion.  Such a technology would allow patients with intractable back pain from end-stage arthritis or scoliosis/kyphosis/spondylolisthesis to get the fusion they need to control their pain, but not be exposed to surgery: several hours in the OR, risk for bleeding/infection/paralysis/etc, 2-4 months of post-OP pain, 3-6 months out of work, among other unfortunate realities regarding fusion surgery.  We have already been able to produce small ossicles in ex vivo disc organs by manipulating NP genes that control extracellular phosphate/pyrophosphate levels (https://doi.org/10.1038/s41598-021-87665-2), and anterior lumbar fusions by growing bone around the IVD (https://doi.org/10.1038/s41598-022-21208-1 ), but we want to get these two successes to combine, where we get a solid arthrodesis through the disc space.  Obtaining this combination of successes was the intention of the current work, and we learned a lot from the experience, but we need to keep on going with slightly different approaches – perhaps manipulating the Pi/PiP genes and adding chABC?  Stay tuned!

Reviewer 3 Report

This study has some interest, but, at this stage, it is affected by serious flaws.

Specifically:

1) The Title is rather complex and difficult for readers. It must be totally amended/changed. In addition, in it the use of Acronyms is excessive: chABC is not well known, therefore Chondroitinase ABC enzyme must be written extensively (then in the text after the first time the Acronym in brackets). The Authors must write an attractive and easier Title (why write in in "not VEGF gene-delivery? and repeat then gene-delivery in the next line?

2) The Introduction contains, as the whole manuscript, too many citations: it reaches now the number of 89! This is not acceptable. Some are put together but for example they are 17-27 that are 11 citations! Maybe 1 or 2 are enough! Most of them unnecessary: so they must be drastically reduced to less that a half in total choosing only the right ones and with better significance for the paper.

3) Materials  and Methods are badly written and not divided into subheadings as for the Results. The Authors must pay a lot of attention in writing this part of the manuscript, make understandable all the passages and experiments that they have performed in a subsequent way.

4) The Figure captions are too long. In Figure 4 the first 3 para from BMP to immunostain must be deleted and reported in the figure: this will avoid the reader to read all the time what any raw and column represents. The Tables 1 and 2 must be transformed in Histograms or something of similar avoiding the numbers in Tables.

5) The Discussion must be better focused on the results/topic and all repetitive or not useful sentences deleted. As in the Introduction the Citations are excessive and they must be strongly reduced to the main ones (essentials). This is a research article and NOT a Review therefore 193 citations are too much: also a reduction to 50% can be not enough.

6) Try to delete or transform the descriptive paragraphs and at the end of the Discussion separate into two different subheadings the Conclusions with the perspectives and the Limitations of this study.

Moderate editing is required. Some sentences/words/verbs are use for colloquial English than for Scientific English.

Author Response

Reviewer #3

Summary –

This study has some interest, but, at this stage, it is affected by serious flaws.

Specifically:

Points –

  • The Title is rather complex and difficult for readers. It must be totally amended/changed. In addition, in it the use of Acronyms is excessive: chABC is not well known, therefore Chondroitinase ABC enzyme must be written extensively (then in the text after the first time the Acronym in brackets). The Authors must write an attractive and easier Title (why write in in "not VEGF gene-delivery? and repeat then gene-delivery in the next line?

The Title was modified to a more easily read and understood description of our results and the model we used.

“Induction of Nucleus Pulposus Angiogenesis by Chondroitinase ABC Treatment: Assessment in a Rodent Anterior Spine Fusion Model”

We agree that chondroitinase ABC is not a very common molecule.  In each paragraph or caption where chondroitinase ABC was mentioned, it is spelled out and the abbreviation in brackets was added.  This appears as yellow or green highlighting.

  • The Introduction contains, as the whole manuscript, too many citations: it reaches now the number of 89! This is not acceptable. Some are put together but for example they are 17-27 that are 11 citations! Maybe 1 or 2 are enough! Most of them unnecessary: so they must be drastically reduced to less than a half in total choosing only the right ones and with better significance for the paper.

In an attempt to not have the manuscript seem to not be supported well enough by the literature, I agree, we went a bit too far in the other direction (over citing).  We made a concerted effort to find the key references to make out points, and to get rid of less important references, or those that were mentioned once only.

The Introduction started with 89 references, and now has 43.  This is a 51.7% reduction.

The entire manuscript had 193 references, and now has 96.   This is a 50.3% reduction.

  • Materials  and Methods are badly written and not divided into subheadings as for the Results. The Authors must pay a lot of attention in writing this part of the manuscript, make understandable all the passages and experiments that they have performed in a subsequent way.

The M&M was divided into subheadings, was re-organized and special attention was made to be certain that it reads logically and explains how another investigator would be able to repeat the experiments, should they want to do so.  There were a multitude of changes, and at least most of these were put into yellow and green highlighted text.

  • The Figure captions are too long. In Figure 4 the first 3 para from BMP to immunostain must be deleted and reported in the figure: this will avoid the reader to read all the time what any raw and column represents. The Tables 1 and 2 must be transformed in Histograms or something of similar avoiding the numbers in Tables.

The captions were all revised and shortened.  Figure 4 (the new Figure 7) was re-composed, labels added to the image itself (and groups added per the request of Reviewer #2).  Tables #1 and #2 were re-configured into the new Figures 4, 5 & 6.

  • The Discussion must be better focused on the results/topic and all repetitive or not useful sentences deleted. As in the Introduction the Citations are excessive and they must be strongly reduced to the main ones (essentials). This is a research article and NOT a Review therefore 193 citations are too much: also a reduction to 50% can be not enough.

The Discussion was re-organized, re-worded and shortened. The text was modified to be more reflective of our major themes (angiogenesis, how chABC affects sFlt, the fibrocartilage healing response, antagonism between chABC and BMP, and that VEGF did not augment BMP), and ability to be read relatively easily.

The Introduction started with 89 references, and now has 43.  This is a 51.7% reduction.

The entire manuscript had 193 references, and now has 96.   This is a 50.3% reduction.

  • Try to delete or transform the descriptive paragraphs and at the end of the Discussion separate into two different subheadings the Conclusions with the perspectives and the Limitations of this study.

The last paragraphs at the end of the Discussion were turned into a limitations paragraph to end the Discussion, and a Conclusions section (one medium and one short paragraph).

Round 2

Reviewer 1 Report

No further comments

No further comments

Author Response

Reviewer #1

Comments and Suggestions for Authors:  No further comments

Reply - We thank you for your comments, they made our manuscript stronger.

Comments on the Quality of English Language:  No further comments

Reply - Thank you.

Reviewer 2 Report

Thanks for addressing previous issues.

Author Response

Reviewer #2

Comments and Suggestions for Authors:  Thanks for addressing previous issues.

Reply - We thank you for your comments.  You made our manuscript stronger!

Reviewer 3 Report

The Authors have amended the manuscript following the majority of previous concerns but it still presets some pitfalls and above all the Authors over write everything even the subheadings: they must be concise and totally change their way to write: they are verbs, prolix. In scientific papers it is needed to be focused and synthetic.

Therefore and for a mere example, the Authors have changed the Title that is still difficult to be understood: be more synthetic and focused.

Moreover the subheadings added in the Results are too long, while they must be concise or they must be further subdivided. The subheadings of the results are correctly written, because they descrive in a few words the Results,  while in the Materials and Methods they must be clear and every experiment must be described in details while the subheading must describe in a few words the content. For example they write: "Bovine and rabbit NP tissue was subjected to Wester blotting to detect sFt." It should be "Western Blotting experiments" or "Western Blotting experiments to detect sFt".

They write: "Statistical data testing, data interpretation and data presentation"while they should write "Statistical evaluation and data analyses" or "Statistics and data analyses".

They write: "Primary tissue culture of bovine nucleus pulpous cells and rabbit disc organs were completed": this sentence is wrong with English mistakes and unclear. They should have written: Bovine nucleus pulpous and rabbit disc organs coltures (with o and not u).

The same is for ALL the subheadings that are wrongly written.

Hope the Authors will find an English speaker expert in writing scientific papers.

The figures of cells in colures are lacking: they must add them before and after the colture completion.

Finally the Limitations: as this Reviewers wrote they must write the Limitations of this study before the conclusions or after them in a separate subheading being clear.

Several mistakes are still present and a detailed checking in the whole manuscript is needed.

Author Response

Reviewer #3

  • The Authors have amended the manuscript following the majority of previous concerns but it still presets some pitfalls and above all the Authors over write everything even the subheadings: they must be concise and totally change their way to write: they are verbs, prolix. In scientific papers it is needed to be focused and synthetic.

Reply – Thank you for this advice.

  • Therefore and for a mere example, the Authors have changed the Title that is still difficult to be understood: be more synthetic and focused.

Reply – The title was altered to be more concise. The most significant finding in the report is the angiogenesis, and so, angiogenesis was the focal point of the title.

It now reads: “Inducing Angiogenesis in the Nucleus Pulposus”

  • Moreover the subheadings added in the Results are too long, while they must be concise or they must be further subdivided. The subheadings of the results are correctly written, because they descrive in a few words the Results, while in the Materials and Methods they must be clear and every experiment must be described in details while the subheading must describe in a few words the content. For example they write: "Bovine and rabbit NP tissue was subjected to Wester blotting to detect sFt." It should be "Western Blotting experiments" or "Western Blotting experiments to detect sFt".

Reply – Thank you for this feedback.  The Subheadings were changed, and now are much more terse and descriptive.  Most of them are 5 words, but range from 3 to 6.

From your Example, the new subheading is:

“2.6 Western blotting experiments”

  • They write: "Statistical data testing, data interpretation and data presentation"while they should write "Statistical evaluation and data analyses" or "Statistics and data analyses".

Reply – We agree, thank you for this feedback.

The new subheading is:

“2.7 Statistical testing and analysis”

  • They write: "Primary tissue culture of bovine nucleus pulpous cells and rabbit disc organs were completed": this sentence is wrong with English mistakes and unclear. They should have written: Bovine nucleus pulpous and rabbit disc organs coltures (with o and not u).

Reply - We agree, thank you for this feedback.

The new subheading is:

“2.5 NP cell and disc organ preparation”

  • The same is for ALL the subheadings that are wrongly written.

Reply - We agree, thank you for this feedback.

The other new subheadings are:

“2.1 Preparing Treatments and animal surgeries

2.2 Noninvasive induced angular displacement assessments

2.3 Bone formation and fusion assessments

2.4 Micro-CT/histology and biomechanics assessments“

  • Hope the Authors will find an English speaker expert in writing scientific papers.

Reply – Thank you for this suggestion, we will keep an eye out for one.

  • The figures of cells in colures are lacking: they must add them before and after the colture completion.

Reply – Using the Reviewer’s description of coltures (in Point #5, above) as a way to better understand which of the cells that were to be shown in colture, we concluded that Reviewer #3 wished to see the bovine NP cells and/or the rabbit disc organs. The bovine NP cells demonstrate no visible change in morphology after being coltured up to passage 10, and experiments were typically done on cells at passage 2-5.  We have generated a new Supplemental Figure A2, with passage 0 cells (newly harvested from bovine tail and attached to a tissue culture dish prior to the first passage) and passage 10 cells (which could have been used for the experiment).

As regards rabbit disc organs: the rabbit disc organs were opened, NP tissue was removed, treated (chondroitinase ABC or Collagenase Type II), and specimens for Western blotting were directly obtained.  No colturing was performed.  No pictures of colturing are available.

  • Finally the Limitations: as this Reviewers wrote they must write the Limitations of this study before the conclusions or after them in a separate subheading being clear.

Reply – A subheading was generated for Limitations, and the section was reworked/reworded to be more concise.